# FGSC: Fuzzy Guided Scale Choice SSD Model for Edge AI Design on Real-Time Vehicle Detection and Class Counting

**DOI:** 10.3390/s21217399

**Published:** 2021-11-07

**Authors:** Ming-Hwa Sheu, S. M. Salahuddin Morsalin, Jia-Xiang Zheng, Shih-Chang Hsia, Cheng-Jian Lin, Chuan-Yu Chang

**Affiliations:** 1Department of Electronic Engineering, National Yunlin University of Science and Technology, Douliu 64002, Taiwan; sheumh@yuntech.edu.tw (M.-H.S.); jamesb09111@gmail.com (J.-X.Z.); hsia@yuntech.edu.tw (S.-C.H.); 2Department of Computer Science and Information Engineering, National Chin-Yi University of Technology, Taichung 411030, Taiwan; cjlin@ncut.edu.tw; 3Computer Science and Information Engineering, National Yunlin University of Science and Technology, Douliu 64002, Taiwan; chuanyu@yuntech.edu.tw

**Keywords:** fuzzy guided scale choice, fuzzy sigmoid function, vehicle detection, fuzzy logic, vehicle class counting, and intelligent AIoT vehicles application

## Abstract

The aim of this paper is to distinguish the vehicle detection and count the class number in each classification from the inputs. We proposed the use of Fuzzy Guided Scale Choice (FGSC)-based SSD deep neural network architecture for vehicle detection and class counting with parameter optimization. The ‘FGSC’ blocks are integrated into the convolutional layers of the model, which emphasize essential features while ignoring less important ones that are not significant for the operation. We created the passing detection lines and class counting windows and connected them with the proposed FGSC-SSD deep neural network model. The ‘FGSC’ blocks in the convolution layer emphasize essential features and find out unnecessary features by using the scale choice method at the training stage and eliminate that significant speedup of the model. In addition, FGSC blocks avoided many unusable parameters in the saturation interval and improved the performance efficiency. In addition, the Fuzzy Sigmoid Function (FSF) increases the activation interval through fuzzy logic. While performing operations, the FGSC-SSD model reduces the computational complexity of convolutional layers and their parameters. As a result, the model tested Frames Per Second (FPS) on edge artificial intelligence (AI) and reached a real-time processing speed of 38.4 and an accuracy rate of more than 94%. Therefore, this work might be considered an improvement to the traffic monitoring approach by using edge AI applications.

## 1. Introduction

Vehicle detection, classification, counting, and monitoring all are regular tasks for the traffic management authority. The volume of vehicles on the roads has an economic impact on the transportation sector and the specific region around the country [1]. Geomagnetic simulation, radio-frequency detection, or vision-based technology are all used to monitor vehicles. Geomagnetic simulation [2] needs to be built beneath the road’s surface, and its execution is quite complicated and expensive. The radio-frequency detection method [3,4] has a good effect on results. In addition, the radio-frequency detection method is suitable for vehicle counting and highway toll ticketing systems in favor of the e-Tag. However, the approach requires an elevated bridge, which is not always accessible everywhere, making it costly and inconvenient for the vendors. However, the image detection approach [5,6,7] is simple to set up and maintain in any location. In addition, the image detection method also can be directly combined with road surveillance cameras. Since there are many security cameras on highways, roads, and streets nowadays, image detection can quickly be related to the existing system.

Computer vision systems extract the salient regions from the background of an input image [8]. Many existing CNN-based salient object identification techniques aim to learn feature vectors at various scales of picture portions, inferring the saliency of each part in the image [9]. The necessity of many labeled images as data in the training stage is a fundamental challenge in the existing deep saliency detection algorithms. The training processes for pixel-wise ground-truth annotation techniques, on the other hand, are time-consuming [10]. Salient Object Detection (SOD) methods expand the object detection, segmentation, and annotating of the training data [11] without human efforts. Although SOD methods provide saliency maps of object areas as output, detecting tiny objects and achieving exact location remains problematic [12]. The SSD [13] model improves default box output space, varied aspect ratios, feature map placement, and the modification of default and bounding boxes to better fit object appearances. SSD has achieved significant progress in detecting small-sized objects and increased localization accuracy with multiscale feature maps and the default boxes method. We adopted the SSD model and extended it with the fuzzy guided scale choice block.

Block sparse RSPCA [5] is useful for vehicle detection and counting. Using digital signal processing-based image recognition technology, the total number of vehicles on the road can be counted. The authors of [6] proposed vehicle counting with real-time proceeding speeds, but their approach used the background subtraction method of low-rank sparse to detect foreground moving objects and counting vehicles. We have considered an AI-based method for vehicle object detection and class counting. Fast-SSD [7] is offered for vehicle identification and counting, and it recognizes objects using six-scale feature maps, resulting in good accuracy. However, Fast-SSD can operate only on one-way roads. Furthermore, the SSD model is a sequential convolution approach, which causes many weights to become zero, wasting time and resulting in low efficiency [14]. As a result, we proposed the FGSC-SSD model for forward passing vehicle detection and backward passing vehicle detection, separately, and class counting. The FGSC-SSD model solves the efficient loss of multiple convolutional layers by utilizing learning weights to impose important and irrelevant features. The operation choice mechanism skips the unimportant characteristics that would have robust effects on the operation speed during the test stage. The main contributions of this work are summarized as follows:❖We set up the Region of Interest (ROI) for quick vehicle detection and class counting from forward passing and backward passing lanes. ❖Proposed FGSC blocks distinguish between significant and irrelevant features. Next, improved system operation speed by skipping unnecessary characteristics.❖We have developed a fuzzy sigmoid function for controlling the activation interval and avoiding saturated output feature values.❖In comparison to the SSD model, the proposed FGSC-SSD model has achieved higher speed under the same detection accuracy for the PASCAL VOC dataset and Benchmark BIT Vehicle dataset.

The FGSC-SSD model-based vehicle detection and class counting system has been deployed on edge AI and in real-time. Its processed speed is about 38.4 FPS for real video. The total accuracy rates achieved for cars, buses, and trucks was 95.1%, 92.3%, and 90.9%, accordingly.

## 2. Related Works

Object detection is more challenging for visual attention, drawing the bounding box around each object of interest and assigning them a class. The deep learning (DL) algorithm has made tremendous progress in object identification, demonstrating, classification, higher feature extraction, and considerably improving detection. One of the most successful applications is the convolutional neural network (CNN) AlexNet [15], which outperformed prior methods.

In the research of object tracking, the two-stage object detection deep neural network models are SPPNet [16], R-CNN [17], Fast R-CNN [18], Faster R-CNN [19], Mask R-CNN [20], ME R-CNN [21], MFR-CNN [22], SWAE [23], and A CoupleNet [24], which all improved performance. The models of the two stages need a Region Proposal Network (RPN) to extract bounding boxes. Those bounding boxes are like ground-truth objects which can precisely locate the object position with good performance. While two-stage solves the instance segmentation problem, it also adds to the network’s processing cost and computational complexity.

For considering the speed performance of the deep learning algorithm, some one-stage methods such as, for example, YOLO [25], SSD [26], Retina-Net [27], SqueezeDet [28]. CornerNet [29], MSA-DNN [30], Image base [31], and DF-SSD [32], represent the object detection deep neural network models. Accuracy and speed are frequently incompatible, as can be seen when the accuracy of feature descriptors is significantly lower than that of a DL technique. The one-stage eliminates the RPN and ROI pooling, so the one-stage approach is faster than two-stage.

Table 1 depicts the performance of the object detection deep neural network models ME R-CNN, MFR-CNN, A-Couple-Net, and DF-SSD on the GPU platform, with lower FPS. The MFR-CNN projected to combine the multi-scale features and the global features improved the accuracy, but the lowest FPS was only 6.9. The DF-SSD tried to combine the shallow and deep characteristics to solve the SSD problem. The DF-SSD model achieved good accuracy for small objects, but the FPS was still low because of the lack of relation between shallow and deep features. The MSA-DNN proposed to combine MSA-DNN and MSAM and extract more features, which improved the accuracy and the FPS value, but those models cannot achieve real-time operation in edge AI. For the Real-Time application, the GPU performance of the SSD and YOLOv4 model’s FPS is high than others for various vehicle detections, but it is hard to achieve real-time operation on an edge AI platform.

The SSD model adds several feature layers at the end of a base network which predicts offsets in default boxes of different scales and aspect ratios and their associated confidences. The SSD model has multi-scale feature map technology for object detection, whereas the YOLO model has a single-scale feature map. In addition, the YOLO model has been developed by intermediate fully connected layers, whereas the SSD model employs coevolutionary filters for each feature map location. The SSD model detects objects using convolutional default boxes from multiple feature maps and matching strategies. In our study, we adopted the SSD model and extended this idea by applying the FGSC block architecture with the fuzzy sigmoid function to significantly increase the quality of the prediction and performance with an even smaller number of parameters.

## 3. Proposed Vehicle Detection and Class Counting System

The vehicle detection and class counting intelligent technique flow chart diagram in Figure 1 shows the working process in three phases. At first, the ROI setup can locate the position of the forward passing and backward passing objects through the object detection window from the road video as an input. The second step is the proposed FGSC-SSD deep neural network model to identify the vehicles from the current frame. The objects are categories as cars, buses, and trucks. Lastly, the vehicle class is counted based on ROI windows setup and working principle. If the input video does not end, the process becomes repeated, following the vehicle detection and class counting flow chart. If it ends, the steps reach the end and show the result.

### 3.1. ROI Setup

Figure 2 shows the ROI set up for the object detection window. The red and pink color frames represent the forward passing detection line and class counting window. Similarly, lime and dark green color frames are the backward passing detection line and the class counting window, correspondingly. Research [33] on automatic traffic counting systems showed the two-lane road has good performance on vehicle class counting.

The passing detection lines are responsible for checking whether the vehicles pass through on the detection lines or not. Consistently, the class counting windows are accountable for vehicle class counting. In both cases, an object passing and the class counting windows are placed behind the passing detection line.

### 3.2. FGSC-SSD Model Architecture

The continuous convolution operations of deep neural networks can approach the weight values to become zero, which causes many parameters to be invalid and inefficient for the operations. Therefore, the authors proposed the FGSC-SSD model architecture. Figure 3 demonstrated the proposed FGSC-SSD deep neural network model’s architecture. The model belongs to eleven convolutional layers (Conv1 to Conv11), five FGSC blocks (FGSC1 to FGSC5), and six convolutional blocks (Conv12 to Conv17). Each layer and block consist of a different number of channels from input to output levels.

The five FGSC blocks have been incorporated into the convolutional layers of Conv3 to Conv9 in the FGSC-SSD model to reduce the performance loss by segmenting the essential characteristics. However, we did not integrate any FGSC blocks in between the input layer and Conv3 layer because shallow features belong to these layers. All initial features are significant for the experiment. Therefore, we cannot skip any characteristics during the initial operation for the first three convolutional layers. These FGSC blocks ignore the insignificant elements from the process and increase the detection speed at the test phase. Alternatively, the Conv9 to Conv11 layers are the deep convolutional layers and have fewer features that are important for the performance. Therefore, we have not added any FGSC blocks in the deep convolutional layers. The operational function of FGSC blocks depends on the global average pooling, fully connected, fuzzy sigmoid function, and scale choice layer. The design of the FGSC block enhances the convolutional operations and reduces the computational complexity.

We used another six convolutional blocks (Conv12–Convo17) for extracting the default boxes which predict the objects in the detection layer. Hard Negative Mining (HNM) was employed to control the positive and negative sample ratios at 3:1 and avoid the weights being biased toward negative samples with the features varying input sizes. The default box is used with the Ground Truth (GT) box to form the Intersection over Union (IoU). The IoU value for positive samples is less than 0.5, and the background probability is less than 0.3, indicating that the sample is a negative sample that can suppress the circumstance of the negative instances. The object location loss function combines convolutional network prediction and default frame prediction to calculate the object location loss of the GT frame corresponding to the positive sample. The combination of x and y coordinates, width, and height are shown in the results at the detection line.

### 3.3. FGSC Block Operation Function

The FGSC block architecture consists of the global intensity, fuzzy guided, and scale choice; the authors propose the global intensity function by the global average pooling [34], and the fuzzy guided function with the fully connected and fuzzy sigmoid. The fully connected layer combines global average pooling and fuzzy sigmoid function, which is shown in Figure 4. Therein, *I* denotes the block input and *H*, *W*, and *C* depict the input feature maps’ height, width, and channel, respectively. The input feature maps *I* pass through the fuzzy guided scale functions. The fuzzy guided is fully connected with global intensity and the results of the fuzzy guided function, and *I* pass through the scale choice layer. The scale choice layer ensures optimized *Y* output selection for the blocks. The input channel number *C* changes into the *C** channel of the output feature map.

#### 3.3.1. Global Intensity Function

The global average pooling extracts features for the global intensity function and is calculated through the following Equation (1). The global average pooling output features represent by *Z = {z_1_, z_2_, …, z_c_}*. The FGSC block input feature *I* = *{I_c_ (i,j), c = 1,*
*…, C}* where *C* is the channel number, *H* is the height, and *W* is the width of *I_c_* feature maps. The summation of input channel features *I_c_* divided by the product of height and width values, then achieved as the result of the global average value.
(1)$$z_{c}=\frac{1}{H\times W}\overset{H}{\underset{i}{\sum }}\overset{W}{\underset{j}{\sum }}I_{c}\left(i,j\right).\;\;\;c=1,2,\ldots ,C$$

#### 3.3.2. Fuzzy Guided 

The fuzzy guide response to achieve the scale values is ***S***. The fully connected and fuzzy sigmoid functions are proposed by the authors to obtain the output value of S. These scale values execute the important and unimportant features from block input ***I***. The fuzzy sigmoid function depicts the fuzzy guide result, and we calculated the fuzzy guide scale value by the following Equation (2).
(2)$$S=F\_Sigmoid\left(fc\left(W_{1},Z\right)\right)$$

Fully Connected

Each channel of the global intensity and fuzzy sigmoid function combined by fully connected in the fuzzy guide operation for learning the average value of Z. The weights and biases of the fully connected Equation (3) are $W_{1}\in R^{\frac{C}{r}\times C}$, r; to control the memory size and calculation quantity, r is set to 2 in this study and $bias\in R^{C}$.
(3)$$K=fc\left(W_{1},Z\right)=Z*W_{1}+bias$$

Fuzzy Sigmoid Function

We incorporated the α parameter into the fuzzy sigmoid function that is shown in Equation (4), where *K*
*= {*$K_{min}$, *k_2_, k_3_ ….*$\;K_{max}$*}* is the output value of the fully connect. The fully connected output’s maximum and minimum values represent the α parameters. The fuzzy range is defined by the highest and lowest fully connect output values. Each FGSC block achieved output values differently [35]. The fuzzy logic has better adaptability especially suitable for nonlinear function, so the authors announced the fuzzy logic to determine the α parameter of the fuzzy sigmoid function.
(4)$$\;\;\;\;S=F\_Sigmoid\left(K\right)=1/\left(1+e^{-\alpha K}\right)$$

In addition, the fuzzy sigmoid function represents classification decisions explicitly in the form of fuzzy rules. This research has developed a new dimension of object detection with the support of the fuzzy sigmoid function. The fuzzy sigmoid function depicts the simple technology and the shortcomings to control the activation interval in the sigmoid function and uses fuzzy logic to select the best parameter.

In this part, we explained why to add *α* parameters into the fuzzy sigmoid function. The FGSC blocks training algorithm automatically deprives the fuzzy set parameters for the “fuzzy rules” in the fuzzy sigmoid function. The FGSC-SSD deep neural network-based object detection and class counting rules obtained, in symbolic form, facilitate the understanding. The maximum and minimum values are 20.539 and −20.310 for the FGSC block no. 2. Figure 5a shows the sigmoid function for *α* = 1, and the upper and lower limits are 6 and −6;alternatively, Figure 5b represents the fuzzy sigmoid function set the *α* parameter to 0.09 so its upper and lower limits become 60 and −60. Then the active area is between 20.539 and −20.310 will be in the saturation zone, so the gradient does not become ‘0′, and it can effectively update the weight values. The fuzzy sigmoid function uses the α value to control the activation interval and used fuzzy logic to find the appropriate α for effectively activating the parameters to improve performance. 

The Fuzzy Sets for α

The F_Sigmoid function has an asymmetric relationship between *K^i^_max_* and *K^i^_min_* values. To define the fuzzy sets, the maximum and minimum values of fuzzy sigmoid are obtained by fully connected output in the FGSC block. The *K^i^_max_* and *K^i^_min_* values are substitute into the following Equations (5)–(7) to get the non-isometric triangular fuzzy sets [36]. The ‘${\tilde{A}}_{\alpha_{mid}}^{i}$’, ‘${\tilde{A}}_{\alpha_{top}}^{i}$’, and ‘${\tilde{A}}_{\alpha_{bottom}}^{i}$’ are defined by the center, maximum, and minimum values which are obtained through a fuzzy set. After training the PASCAL VOC data set [37], the F_Sigmoid function obtained 13 non-isometric triangular fuzzy sets ${\tilde{A}}_{\alpha }=\left\{{\tilde{A}}_{\alpha }^{i},i=1,\ldots ,13\right\}$ that are shown in Figure 6.
(5)$${\tilde{A}}_{\alpha_{mid}}^{i}=\ln\left(\frac{1}{4}\right)/\left(\frac{\left|K_{max}^{i}\right|+\left|K_{min}^{i}\right|}{2}\right)$$
(6)$${\tilde{A}}_{\alpha_{top}}^{i}=\ln\left(\frac{1}{9}\right)/\left(\frac{\left|K_{max}^{i}\right|+\left|K_{min}^{i}\right|}{2}\right)$$
(7)$$\;\;{\tilde{A}}_{\alpha_{bottom}}^{i}=\ln\left(\frac{3}{7}\right)/\left(\frac{\left|K_{max}^{i}\right|+\left|K_{min}^{i}\right|}{2}\right)$$

For example, after training the FGSC, block 1 has *K^i=1^_max_* = 48.4701, *K^i=1^_min_* = −62.0113. Those values are placed into Equations (5)–(7) and obtain the fuzzy set ‘*fzy2′* mentioned in Figure 6, where ${\tilde{A}}_{\alpha_{mid}}^{i=1}$ = 0.0251, ${\tilde{A}}_{\alpha_{top}}^{i=1}=0.0398$ and ${\tilde{A}}_{\alpha_{b.tom}}^{i=1}$ = 0.0153.

2.The Fuzzy Sets for $K_{max}$ and $K_{min}$

The following Figure 7 and Figure 8 are the non-isometric triangular maximum and minimum fuzzy sets ${\tilde{A}}_{K_{max}}=\left\{{\tilde{A}}_{K_{max}}^{i},i=1,\ldots ,13\right\}$ and ${\tilde{A}}_{K_{min}}=\left\{{\tilde{A}}_{K_{min}}^{i},i=1,\ldots ,13\right\}$ for the *K_max_* and, *K_min_* values, respectively. According to the previous ${\tilde{A}}_{\alpha_{mid}}^{i}$ value, the authors attempted to construct the *K_max_* and, *K_min_* fuzzy sets based on the F_Sigmoid function. The ‘${\tilde{A}}_{K_{\max\_mid}}^{i}$’ is the absolute average value for *K^i^_max_* and *K^i^_min_* that is achieved by the following Equation (8). The ‘${\tilde{A}}_{K_{\max\_top}}^{i}$’ is the top value of fuzzy set for ‘${\tilde{A}}_{K_{max}}$’ which is calculated by the Equation (9). The ‘${\tilde{A}}_{K_{\max\_b.tom}}^{i}$’ is the bottom value for the fuzzy set of ‘${\tilde{A}}_{K_{max}}$’ that is calculated through Equation (10).
(8)$${\tilde{A}}_{K_{\max\_mid}}^{i}=\frac{\left|K_{max}^{i}\right|+\left|K_{min}^{i}\right|}{2}$$
(9)$${\tilde{\;A}}_{K_{\max\_top}}^{i}=-\ln\left(\frac{1}{9}\right)/\alpha_{mid}^{i}$$
(10)$${\tilde{A}}_{K_{\max\_bottom}}^{i}=-\ln\left(\frac{3}{7}\right)/\alpha_{mid}^{i}$$

Similarly, in the FGSC block 1, the *K^i=1^_max_* and *K^i=1^_min_* values were substituted into Equations (8)–(10) and obtained a maximum fuzzy set “*b12*” which is mentioned in Figure 7. The fuzzy set center value is ‘${\tilde{A}}_{K_{\max\_mid}}^{i}$’ = 55.2407, the top value ‘${\tilde{\;A}}_{K_{\max\_top}}^{i}$’ = 87.5388, and the bottom value ‘${\tilde{A}}_{K_{\max\_b.tom}}^{i}$’ = 33.7568, respectively.

Next, using Equations (11)–(13), a minimum fuzzy set ‘*s12′* was obtained which is mentioned in Figure 8, where the fuzzy set-top value is ‘${\tilde{\;A}}_{K_{\min\_top}}^{i}$’ = −33.7568, the bottom value ‘${\tilde{A}}_{K_{\min\_b.tom}}^{i}$’ = −87.5388 and the center absolute value ‘${\tilde{A}}_{K_{\min\_mid}}^{i}$’ = 55.2407, respectively. 

The symmetric relationship between maximum and minimum fuzzy sets of F_Sigmoid function is shown through Equations (11)–(13). The ‘${\tilde{A}}_{K_{\min\_mid}}^{i}$’ is the center position of the fuzzy set ‘${\tilde{A}}_{K_{\min}}$’, ‘${\tilde{A}}_{K_{\min\_top}}^{i}$’ is the top position of the fuzzy set, and ‘${\tilde{A}}_{K_{\min}}$’ and ‘${\tilde{A}}_{K_{\min\_b.tom}}^{i}$’ is the bottom position for the fuzzy set of ‘${\tilde{A}}_{K_{\min}}$’, respectively.
(11)$${\tilde{A}}_{K_{\min\_mid}}^{i}=-{\tilde{A}}_{K_{\max\_mid}}^{i}$$
(12)$${\tilde{A}}_{K_{\min\_top}}^{i}=-{\tilde{A}}_{K_{\max\_b.tom}}^{i}$$
(13)$${\tilde{A}}_{K_{\min\_b.tom}}^{i}=-{\tilde{A}}_{K_{\max\_top}}^{i}$$

3.The Fuzzy Logic Processing

A fuzzy set assigns a membership grade, which selects an integer number from the interval of (0, 1). From the context of fuzzy sets framework emerges fuzzy logic which may address computational perception and cognition-related information: information that is unclear, imprecise, incomplete, or without sharp boundaries. In the development of intelligence, decision-making, identification, pattern recognition, optimization, and control systems, approaches based on fuzzy logic are available [38]. We have utilized fuzzy logic to select the best parameters during the controlling fuzzy sigmoid function activation interval. The fuzzy logic inference block diagram is shown in Figure 9, which consists of Fuzzification, Rulesets, Inference, and Defuzzification. The non-isometric triangular fuzzy sets are used to signify the relationship between input and output of fuzzy logic inference. Fuzzification, Inference, Rulesets, and Defuzzification are the four separate functions of the inference process, where the *K_max_* and *K_min_* are the inputs of fuzzification. The fuzzification mainly classifies the input values for corresponding fuzzy sets. The rule sets store all the regulations that are obtained by experiment. The inference combines the fuzzy sets and rules from Table 2 to determine the fuzzy *α* value. Then, defuzzification converts the fuzzy α value to the crisp α value (α_crisp_) for the fuzzy sigmoid function.

4.Fuzzification

The fuzzification mainly processes fuzzy information. Firstly, the fuzzy sets are defined for the inputs of $K_{max}$ and $K_{min}$. The fuzzy information is calculated through membership grade [39], and the membership grade of x is denoted by µ(x). In addition, the membership grade for $K_{max}$ and $K_{min}$ in fuzzy logic which can be calculated by the functional Equations of (14) and (15). The membership grade of fuzzy sets for the $K_{max}$ is denoted by the $\mu_{{\tilde{A}}_{K_{\max}}^{i}}\left(K_{max}\right)$ and $K_{min}$ is denoted by $\mu_{{\tilde{A}}_{K_{\min}}^{i}}\left(K_{min}\right)$. $\tilde{A}$ = Fuzzy set, and $\mu_{\tilde{A}}\left(x\right)$ = Membership grade.
(14)$$\mu_{{\tilde{A}}_{K_{\max}}^{i}}\left(K_{max}\right)=\{\begin{array}{cc} \frac{{\tilde{A}}_{K_{\max\_top}}^{i}-K_{max}}{{\tilde{A}}_{K_{\max\_top}}^{i}-{\tilde{A}}_{K_{\max\_mid}}^{i}} & {\tilde{A}}_{K_{\max\_mid}}^{i}<K_{max}\leq {\tilde{A}}_{K_{\max\_top}}^{i}\;\\ 1 & {\tilde{A}}_{K_{\max\_mid}}^{i}=K_{max}\\ \frac{K_{max}-{\tilde{A}}_{K_{\max\_b.tom}}^{i}}{{\tilde{A}}_{K_{\max\_mid}}^{i}-{\tilde{A}}_{K_{\max\_b.tom}}^{i}} & {\tilde{A}}_{K_{\max\_b.tom}}^{i}\leq K_{max}<{\tilde{A}}_{K_{\max\_mid}}^{i} \end{array}$$
(15)$$\mu_{{\tilde{A}}_{K_{\min}}^{i}}\left(K_{min}\right)=\{\begin{array}{cc} \frac{K_{min}-{\tilde{A}}_{K_{min\_top}}^{i}}{{\tilde{A}}_{K_{min\_mid}}^{i}-{\tilde{A}}_{K_{min\_top}}^{i}} & {\tilde{A}}_{K_{min\_mid}}^{i}<K_{min}\leq {\tilde{A}}_{K_{min\_top}}^{i}\;\\ 1 & {\tilde{A}}_{K_{min\_mid}}^{i}=K_{min}\\ \frac{{\tilde{A}}_{K_{min\_b.tom}}^{i}-K_{min}}{{\tilde{A}}_{K_{min\_b.tom}}^{i}-{\tilde{A}}_{K_{min\_mid}}^{i}} & {\tilde{A}}_{K_{min\_b.tom}}^{i}\leq K_{min}<{\tilde{A}}_{K_{min\_mid}}^{i} \end{array}$$

If the maximum fully connected value of $K_{max}$ = 15.108 and minimum fully connected value of $K_{min}$ = −12.153, then Figure 10 depicts the membership grade calculation.

5.Rulesets 

The rules of fuzzy sets in Equation (16) have been customized by the experimental results then stored at inference. For the proceeding, “If” expresses antecedent and “Then” represents the consequent, whereas ‘*bi’* and ‘*sj*’ are the sequences of maximum and the minimum fuzzy sets. The ‘${\tilde{A}}_{K_{max}}^{bi}$’ are the 13 maximum fuzzy sets represented by ‘*b1* to *b13′* and ‘${\tilde{A}}_{K_{min}}^{si}$’ are the 13 minimum fuzzy sets demonstrated by ‘*s1* to *s13′*, which represent a total of 169 rules that are shown in Table 2. The fuzzy set column ‘*b13′* and the row ‘*s13′* belongs to all ‘*fzy1′* fuzzy logic rules. On the contrary, the fuzzy set ‘*b1′* and ‘*s1′* only belong to ‘*fzy13′* logic. The rules table expresses a diagonal symmetric relationship between ‘${\tilde{A}}_{K_{max}}^{bi}$’ maximum fuzzy sets and ‘${\tilde{A}}_{K_{min}}^{sj}$’ minimum fuzzy sets.

**Rule:** If $K_{max}$ belongs to ${\tilde{A}}_{K_{max}}^{bi}$ and $K_{min}$ belongs to ${\tilde{A}}_{K_{min}}^{sj}$ then α_crips belong to ${\tilde{A}}_{\alpha }^{bi,sj}$.
*i*, *j* = 1, 2, ……, 13(16)

6.Inference Process

The Inference combines the fuzzification outputs and rules to find the fuzzy α value. The Momani inference method [39] has been followed to obtain the antecedent and membership grade by the following Equations (17) and (18). The maximum and minimum membership grades are denoted by ($\mu_{A_{K_{max}}}^{bi}\left(K_{max}\right)$ and $\mu_{A_{K_{min}}}^{sj}\left(K_{min}\right)$). The two inputs do ‘min’ value to obtain the result of the antecedent (${\tilde{C}}^{bi,\;si}$), then ${\tilde{C}}^{bi,sj}$ do min with $\mu_{\alpha }^{bi,sj}\left(\alpha \right)$ to get the membership grade of the ${\tilde{A}}_{\alpha }^{i}$ on the ith rule. Finally, process the max value with membership grade of the ${\tilde{A}}_{\alpha }^{bi,sj}$ of all rules to obtain $\mu_{{\tilde{A}}_{a}}\left(\alpha \right)$. Figure 11 illustrates the inference process and rules calculation.

$Rule\;1:\;\;\;If\;K_{max}\;is\;b10\;and\;K_{min}\;is$ s10 then α is fzy4

$Rule\;2:\;\;\;If\;K_{max}\;is\;$*b*$\;11\;and\;K_{min}\;is$ s10 then α is fzy3
(17)$${\tilde{C}}^{b,s}=\min\left[\mu_{{\tilde{A}}_{K_{\max}}^{s}}\left(K_{max}\right),\mu_{{\tilde{A}}_{K_{\min}}^{b}}\left(K_{min}\right)\right]$$
(18)$$\mu_{{\tilde{A}}_{a}}\left(\alpha \right)=\max\left[\min\left({\tilde{C}}^{s,b},\mu_{{\tilde{A}}_{\alpha }^{s,b}}\left(\alpha \right)\right)\right]$$

7.Defuzzification

Defuzzification is the conversion procedure of a fuzzy set to transform a fuzzy output into a single crisp value. The defuzzification value in the fuzzy logic controller takes action to control the process. This approach offers a crisp value based on the center of gravity for the fuzzy set. There are several sub-areas inside of the total membership function employed for the combined control operation. The area and center of gravity or center of each sub-area are computed, and all these sub-areas are summed to identify the defuzzification value of a discrete fuzzy set. The inference comes with fuzzy sets that need to find the usable value through defuzzification. We have employed the method of the center of gravity for defuzzification by using Equation (19) where α_crisp is placed into α value for the fuzzy sigmoid function. α_crisp_ is the fuzzy sigmoid parameter, and $\mu_{A_{\alpha }}\left(\alpha \right)$ is the total membership grades of α_crisp_. The defuzzification process and calculation shown in Figure 12.
(19)$$\alpha_{crisp}=\sum^{\;}\alpha \cdot \mu_{{\tilde{A}}_{a}}\left(\alpha \right)/\sum^{\;}\mu_{{\tilde{A}}_{a}}\left(\alpha \right)$$

#### 3.3.3. Scale Choice

We proposed a Scale Choice (SC) layer that has training and testing labels, such as the training label shown in Figure 13. The *I = {I_c_, c= 1, ……., C}* is the inputs of the FGSC block, *S = {S_c_, c = 1, …., C}* is the outputs of F_Sigmoid and *W = {W_c_, c = 1, …, C, 0 ≤ W_c_ ≤ 1}* is the weights of the scale choice. The ***S_C_*** is responsible for recognizing the important channel feature of ***I_C_***. The main operation of S_C_ layers is the input feature I_C_ of each layer which is multiplied by the scale value, and then multiplied by the weight value W_C_. The W_C_ learned the importance of each channel for the entire data set. If the *W_C_ = 1*, it means this channel feature is most important for the entire data set. When the *W_C_ = 0*, it means this channel feature is unimportant for the operation. These operations imposed important channel features through weights learning. The authors distinguished the importance of the operation choice mechanism that can skip the unimportant features on the test stage. The output of the scale choice layer is *Y = {Y_c*_, c* = 1, …., C*}*. The *C** is the remaining channel after the operation choice mechanism.

Operation Choice Mechanism

The scale choice recognizes the significant channel characteristics from the input channel to acquire the activation function scale values. If the *W_C_* value is larger than the Jump-threshold (*thr*) value after training the Scale Choice layer, the feature values are multiplied by *S_C_* and *W_C_*. The ‘*thr’* level was set at 0.2 for this study. The following is the Algorithm 1 for the Scale Choice layer operation mechanism.
**Algorithm 1****Operation choice mechanism.**Input: SAC Block input I={Ic, c=1,…,C},     F_Sigmoid output S={sc, c=1,…,C},Scale Choice layer weights W={wc, c=1,…,C,0≤wC≤1} Output: bth SAC Block output Ycb***1**  Set thr = 0.2**2**    Set cb*=0**3**    for *c* = 1: C do**4**      if wc > thr do**5**        Ycb*=Ic×sc×wc**6**         cb*++ **7**    endend

Otherwise, the small *W_C_* stands for a lower feature value that has little effect on the following up to skip convolution.

Effects of Different Jump-threshold

Table 3 shows the test results of the various Jump-thresholds for PASCAL VOC datasets. The results are varying on the different jump thresholds. The result depicts that the FPS is low when the Jump-threshold is 0.08, and the accuracy is inadequate when the Jump-threshold is 0.3, so an appropriate threshold must look for better performance. Contrary to Jump-threshold 0.1 and 0.3, if the speed increases, the accuracy decreases, and vice versa. If the accuracy increases, the speed decreases. In this situation, the jump-threshold 0.2 has good speed and accuracy. Because we tend to increase the speed, the FGSC-SSD jump threshold is 0.2.

FGSC Block Weights distribution

We analyzed the weights of each FGSC block at the different scale choice layers for the PASCAL VOC datasets. Figure 14, Figure 15, Figure 16, Figure 17 and Figure 18 represents the training weights distribution diagram. The weight values for FGSC1- Scale Choice rarely approached the 0, so the features of the FGSC1 block are most important. To have a clear understanding of the weight distribution of different layers, we set the jump threshold as 0.2 and make the figures on the percentage of skipping weight parameters in different scale choice layers. Table 4 shows the information of average weights and eliminating parameters for all FGSC blocks. The FGSC block 1 leaped 6.2% parameters, and the value of the average weight is 0.49. The FGSC block 5 skipped the 54% parameter, and the average weight value is 0.30. According to test results, the deep block leaps the higher percentage of the parameters and the lower percentage of the average weight value.

## 4. Vehicle Detection and Class Counting Process

### 4.1. Vehicle Detection Process

The operational function of vehicle detection and class counting overall process is illustrated in Figure 19. We set up passing detection lines and class counting windows separately for smooth and quick operation. The passing detection line detects whether the vehicle passes the line or not. If it passes the detection lines and reaches the center point of the class counting window, then the vehicle will be counted in a specific class. The passing detection lines were developed by a Gaussian Mixture Model (GMM) [40], which identifies whether the vehicle arrived and passed the detection line or not. We used background GMM color to build the passing detection line background pixels. When the car passed through the detection line, the pixel’s color changed to the foreground GMM color. In addition, the background pixels set are 0 and the foreground pixels set are 1 so when the detection line GMM color pixels become 1 then the model considered the vehicle as passing the line. After passing the detection lines, when the vehicle reached the center of the class counting window then the algorithm counted the vehicles in particular classes. In Figure 19, the forward passing detection line’s color is still black because there is no vehicle, but the backward passing detection line’s color has changed to the foreground pixels’ color while a vehicle arrived at the detection line and vehicle class was counted. 

### 4.2. Detection Result at Different Weather Condition

To ensure safety issues and robust accurate object detection, the ability to recognize the object in different weather conditions is crucial for the deep learning algorithm. Under challenging weather, particularly rainy days, the performance of object detection algorithms might be reduced considerably. The FGSC-SSD deep neural network model can perform under various weather conditions. For example, in rainy weather, the visual conditions for vehicle detection and class counting experimental result are shown in Figure 20. The prediction accuracy for vehicle detection and class counting process is somewhat changed because of the effect of glare on visibility. Weather conditions include the meteorological changes of the environment due to precipitation including clear weather, rainwater, as well as cloudy weather. The performance of vision-based object detection methods drops significantly under rainy conditions because the camera sensors usually suffer from lowered gradient sizes, which cause the bounding boxes to vary their location and size throughout the detection process. In addition, the categorization values can be reduced, which indicates the increasing uncertainty that can, however, be neglected in near distances. In addition, it could be overcome by collecting a dataset with vehicles in rain scenes and training a state-of-the-art deep learning model using this dataset.

Extreme weather conditions have a significant impact on our everyday lives in a variety of ways. The object visibility is one of them, which is very limited in adverse weather conditions such as fog, ice, snow, and dust. In cold weather conditions, ice, fog, dust, and light snow typically affect visibility. The grayscale image reflects color information to a large context and indirectly shows the feature information. We analyzed the influence of visibility in foggy weather on the accuracy of computer vision obstacles and detection. In this study, we considered visual conditions as the significant changes in the appearance of the experiment during the nighttime effect of brightness on perceptibility. However, the weather conditions are the barometric changes of the environment due to precipitation, including clear and cloudy weather at night. The experimental results during the nighttime with foggy weather show in Figure 21. We have trained the dataset with vehicles in foggy scenes and are training the deep learning model using this dataset.

Machine learning vision technology has an impact on weather conditions. To perform a comparison at different weather conditions, the detection result shows that the nighttime with foggy weather has the lowest accuracy. Whereas the detection result during the rainy day achieved good performance. At the normal weather condition, our proposed FGSC-SSD deep neural network model reached the highest detection accuracy.

## 5. Experiment and Results Comparison

We utilized the PASCAL VOC datasets and the Benchmark BIT Vehicle dataset [41] to evaluate the performance of our proposed FGSC-SSD deep neural network model, trained on the “Caffe” [42] environment. The initial setup for the experiment: learning rate at 0.0001, the momentum as 0.9, and the batch size as 8. For weight initialization, we used a weight initialize strategy and stochastic gradient descent to optimize network weight parameters. To evaluate the performance of proposed FGSC-SSD deep network model in terms of detection, we considered mAP for accuracy and FPS for speed measurement.

### Performance of the Test Datasets

Table 5 displays the performance of various deep neural network models for the PASCAL VOC (07 + 12) datasets. We have considered different GPU platforms to examine the efficiency of those models. On the Nvidia Titan Xp and Titan X platforms, the SSD model has 31.5 and 46 FPS which are better than other ME R-CNN, MSA-DNN, MFR-CNN, DF-SSD, and ACoupleNet approaches. Next, we replaced the platforms with 1080 Ti and Xavier, and the video base framework [43] SSD speed changed to 56 and 16.9, respectively. At the same stages, the YOLOv4 [44] has better precision of 78.9 and YOLOv3 s-highest FPS on 1080 Ti, but the proposed FGSC-SSD achieved the highest inference speed of 64.9 and 21.7 on those platforms, which is higher than the YOLOv3, YOLOv4, and SSD models. Compare to the SSD model, the proposed FGSC-SSD model has the highest accuracy and inference speed which approximates real-time processing speed for edge AI platforms. 

For the benchmark BIT-Vehicle datasets, Table 6 shows the test results of vehicle detection accuracy and inference speed for the YOLOv3, YOLOv4, SSD, and proposed FGSC-SSD deep neural network models on the platform of 1080 Ti and Xavier. Among those models, the YOLOv3 has the highest accuracy (96.3%), whereas the inference speed is second position on both platforms. However, the YOLOv4 and SSD models exhibit an accuracy 95.1 percent and 91.4 percent, respectively, and inference speed is 25.3 and 23.9 on the 1080 Ti platforms. In addition to Xavier platform, the YOLOv4 and SSD models achieved accuracy 95.1 percent and 91.4 percent same as the previous platform, but the FPS is 17.6 and 15.4, respectively. However, the proposed FGSC-SSD deep neural network model has reached highest inference proceeding speed of 26.8 and 21.7 on both platforms with high accuracy (95.5%), which is the second highest position for both platforms. In comparison to the baseline network SSD model, the proposed FGSC-SSD deep neural network model has achieved better accuracy and FPS on both platforms for the benchmark BIT-Vehicle datasets. Considering to the speed, the proposed FGSC-SSD deep neural network model reached the highest inference speed among those models.

Table 7 illustrated the accuracy performance for SSD and FGSC-SSD test results for specific vehicles for both models. Table 8 shows the result for video processing result on Nvidia Xavier. The FGSC-SSD model achieved an FPS of 38.4 which is faster than the backbone SSD model.

## 6. System Implementation

Figure 22 shows vehicle detection and class counting system implementation. We have utilized the edge AI platform for system implementation. In this experiment, the authors used actual mountain road videos for one day as test data. Table 9 depicts the result of vehicle detection and class counting. The detection and class counting accuracy for the car, bus, and truck are 96.7%, 95.5%, and 94.5%, respectively.

The actual results for vehicle detection and class counting from video are shown in Table 9. In total, 61 cars, 44 buses, and 36 trucks went through the vehicle detection lines, with 59 sedans, 42 buses, and 34 trucks properly counted, respectively, with 96.7 percent, 95.5 percent, and 94.5 percent accuracy. However, we noticed some errors in the class counting and reviewed the input data and results again. We found the causes of erroneous counting when two sedans of very old models passed the windows, but their images did not belong to the datasets. In addition, two recondition pickup vans went through detection lines that did not look like trucks. Furthermore, the two buses that crossed both detection lines that made an error for the class counting window.

## 7. Discussion

For considering the higher speed and real-time AIoT applications, we proposed the FGSC-SSD deep neural network model for performing an Edge AI platform. The proposed FGSC-SSD model can accomplish on edge computing for Real-Time IoT applications for vehicle detections. The fuzzy guided algorithm has reduced the errors and improved the vehicle detection rate and class counting. We have utilized a CPU Intel Core i7 8700k, GPU is Nvidia GTX 1080 Ti, CUDA version is 9.0, and cuDNN version is 7.0 to measure the performance by mAP and FPS. For the test experiment, we captured a real video from a mountain road in Taiwan. For future work, we intend to prepare more small-size vehicle images as training datasets to improve the performance capability and accuracy. Additionally, vehicle images datasets from different angles and various weather conditions can boost the performance of the proposed FGSC-SSD deep neural network model for vehicle detection and class counting system for all sorts of vehicles.

## 8. Conclusions

In this paper, the authors proposed an FGSC-SSD deep neural network model for vehicle detection and class counting with passing detection lines and class counting windows. The FGSC blocks comprise the global intensity, fully connected layers, fuzzy sigmoid function, and the scale choice layer. Hence, the fuzzy sigmoid function controls the activation interval and avoids unnecessary features falling into the saturation zone. The weight value is particularly significant for the scale choice layer, as it supports in learning the dataset’s essential feature map and determining how to accomplish the counting optimization impact.

For performance, compared to the baseline network, the proposed FGSC-SSD has achieved higher proceeding speed and accuracy than the SSD model. For the authentic road video tests, vehicle detection and class counting accuracy of cars, busses, and trucks achieved 96.7%, 95.5%, and 94.5% accuracy, respectively. Moreover, the proceeding speed of the proposed FGSC-SSD model achieved 38.4 FPS, which is real-time proceeding speed on the Edge AI platform.

## Figures and Tables

**Figure 1 sensors-21-07399-f001:**
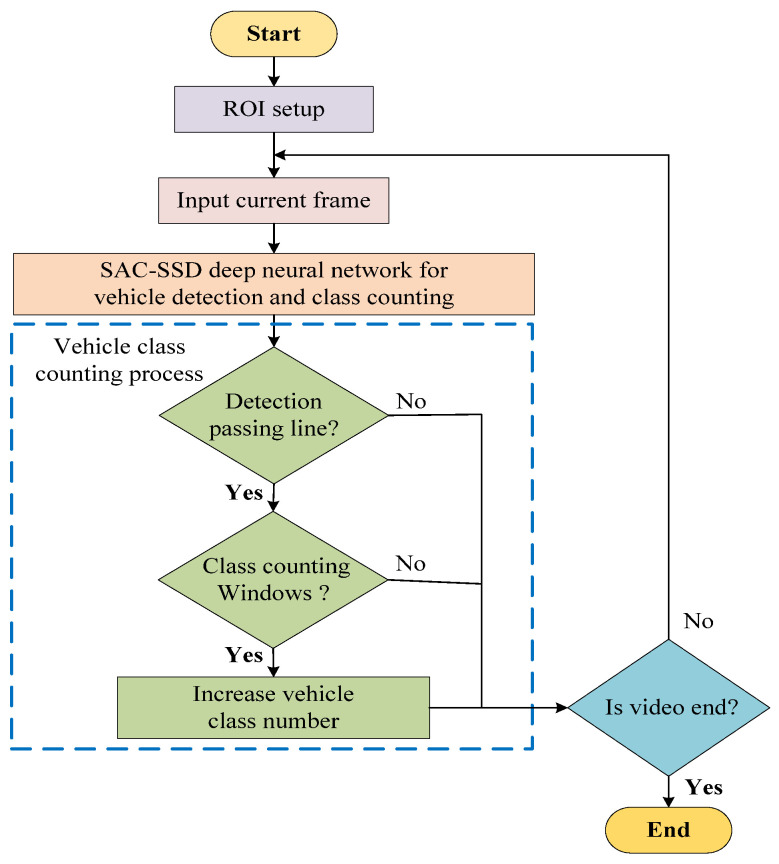
Vehicle detection and class counting flow chart.

**Figure 2 sensors-21-07399-f002:**
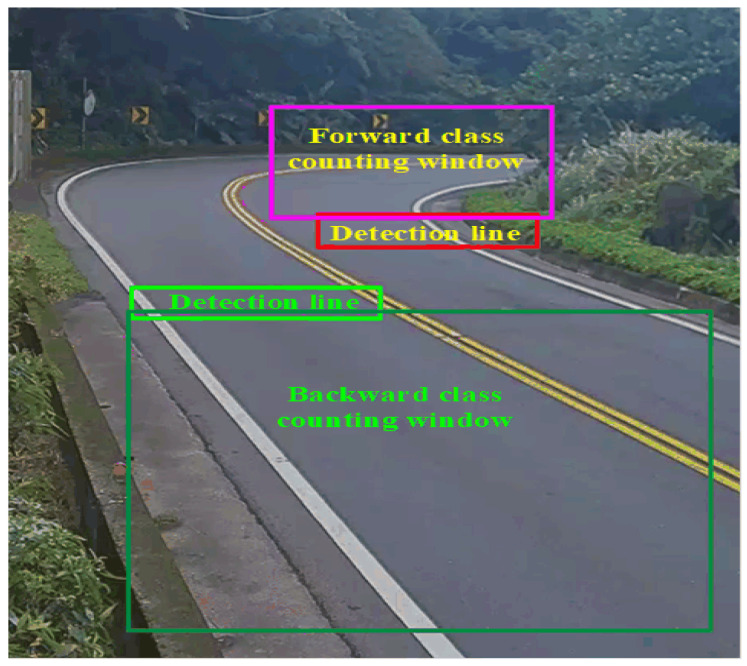
The vehicle’s detection window.

**Figure 3 sensors-21-07399-f003:**
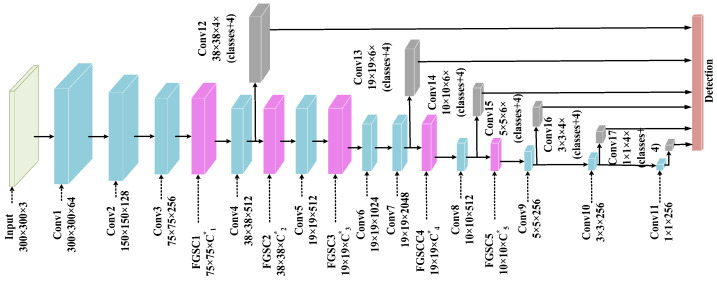
The proposed FGSC-SSD model architecture.

**Figure 4 sensors-21-07399-f004:**
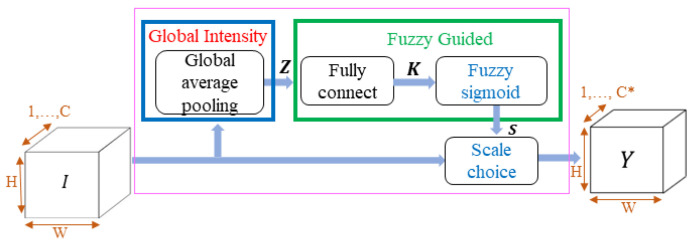
FGSC block architecture.

**Figure 5 sensors-21-07399-f005:**
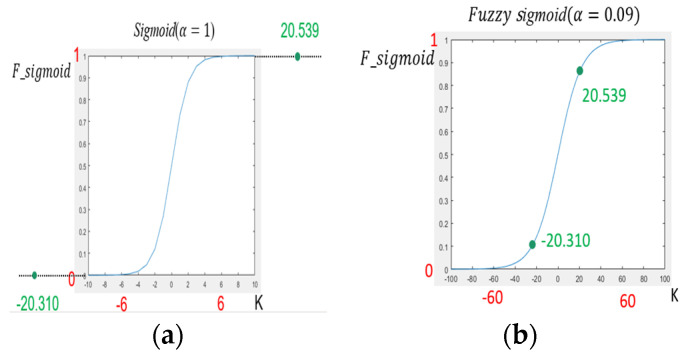
The function of (**a**) Sigmoid function. (**b**) Fuzzy sigmoid function.

**Figure 6 sensors-21-07399-f006:**
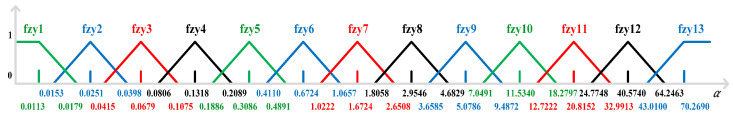
The ${\tilde{A}}_{\alpha }$ fuzzy set for α.

**Figure 7 sensors-21-07399-f007:**
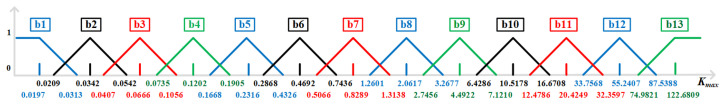
The ${\tilde{A}}_{K_{max}}$ maximum fuzzy set.

**Figure 8 sensors-21-07399-f008:**
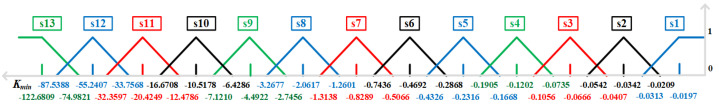
The ${\tilde{A}}_{K_{min}}$ minimum fuzzy set.

**Figure 9 sensors-21-07399-f009:**
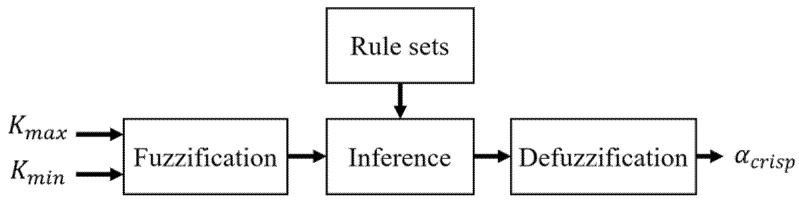
The fuzzy logic inference process.

**Figure 10 sensors-21-07399-f010:**
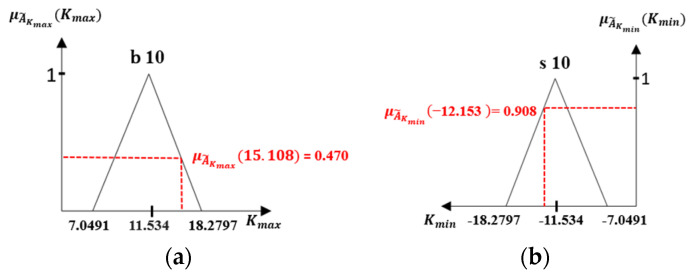
Fuzzification process (**a**) Maximum membership grade (**b**) Minimum membership grade.

**Figure 11 sensors-21-07399-f011:**
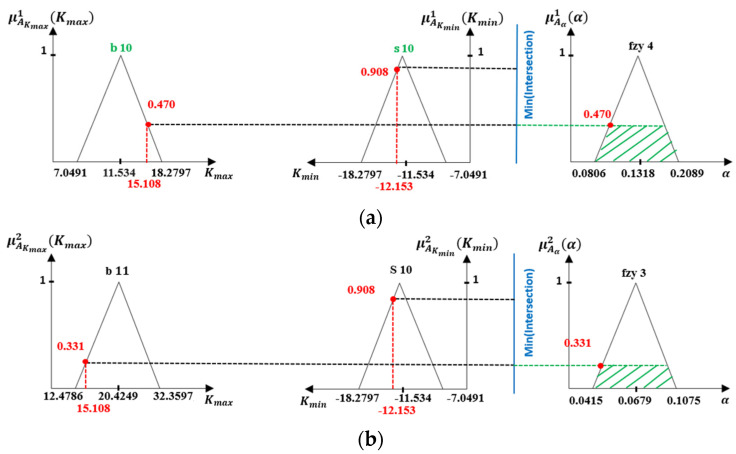
Mamani inference (**a**) Rules1 implementation (**b**) Rules 2 implementation.

**Figure 12 sensors-21-07399-f012:**
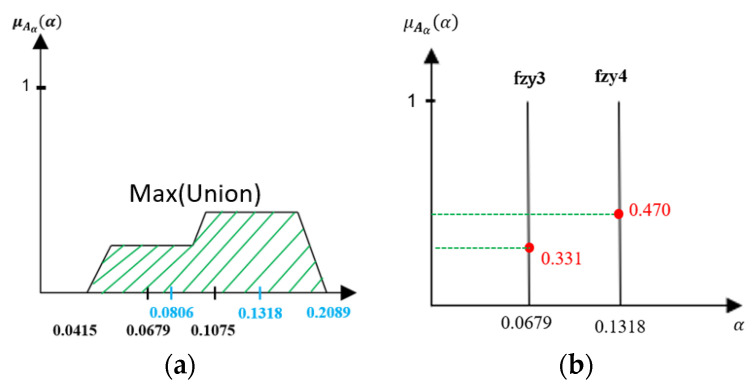
Defuzzification (**a**) Maximum Union of intersection and (**b**) Membership grades calculation.

**Figure 13 sensors-21-07399-f013:**
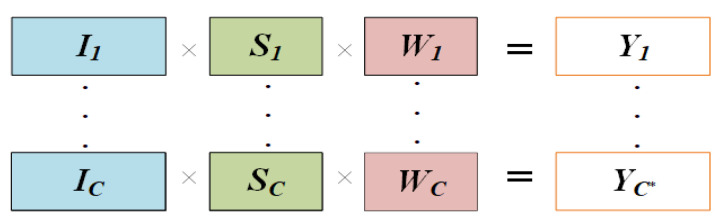
Training mode scale choice layer architecture.

**Figure 14 sensors-21-07399-f014:**
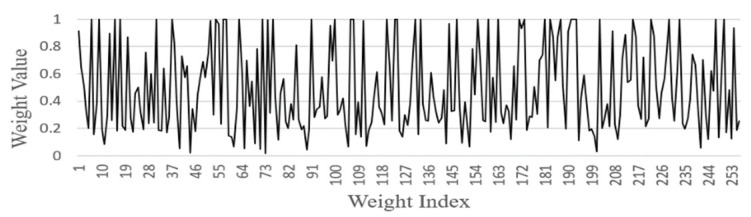
FGSC1-Scale choice layer weight.

**Figure 15 sensors-21-07399-f015:**
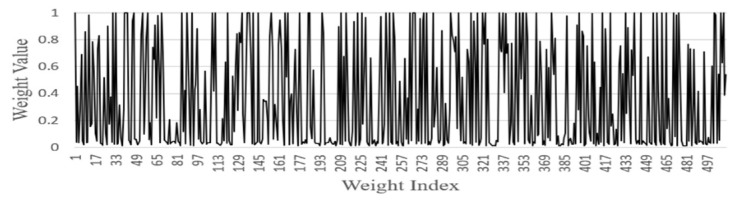
FGSC2-Scale choice layer weight.

**Figure 16 sensors-21-07399-f016:**
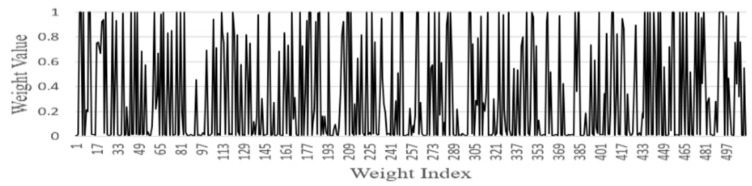
FGSC3-Scale choice layer weight.

**Figure 17 sensors-21-07399-f017:**
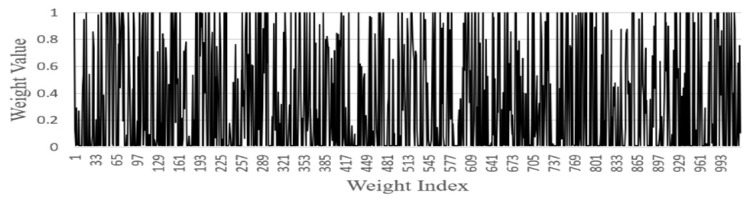
FGSC4-Scale choice layer weight.

**Figure 18 sensors-21-07399-f018:**
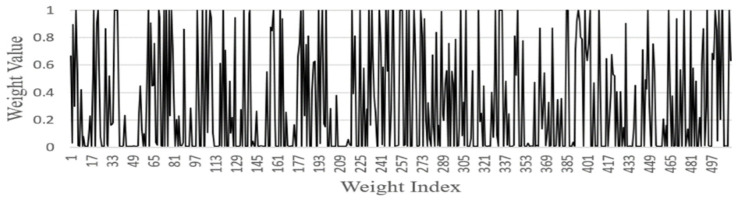
FGSC5-Scale choice layer weight.

**Figure 19 sensors-21-07399-f019:**
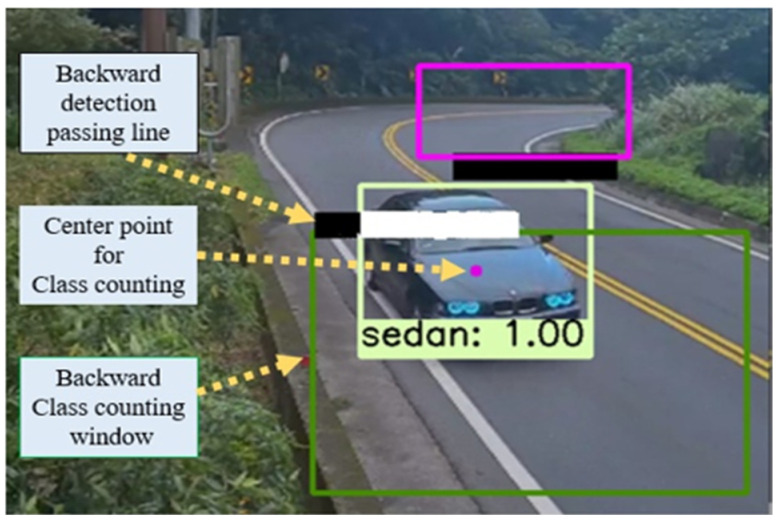
Vehicle detection and class counting process.

**Figure 20 sensors-21-07399-f020:**
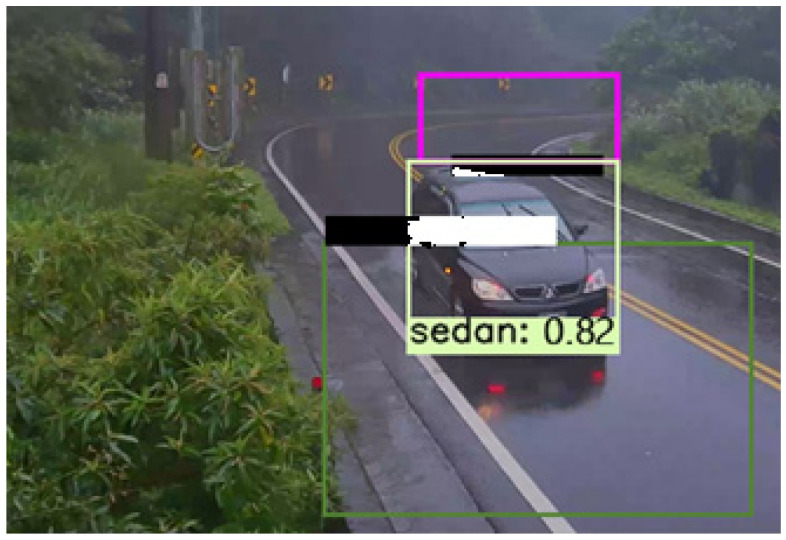
The detection result during the rainy day.

**Figure 21 sensors-21-07399-f021:**
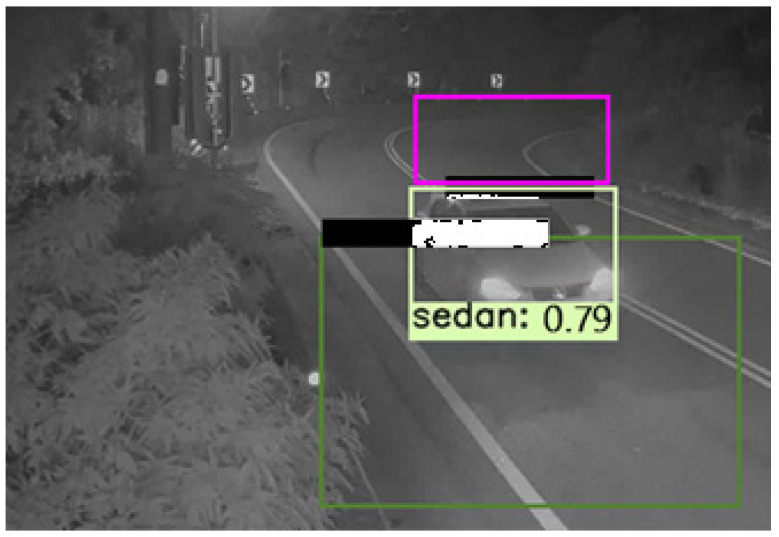
The detection result at nighttime with foggy weather.

**Figure 22 sensors-21-07399-f022:**
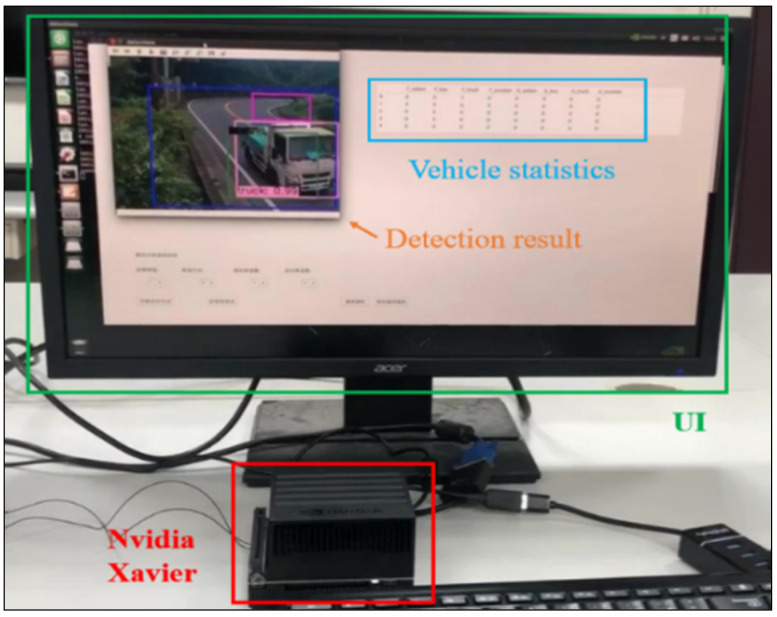
The Edge AI system implementation on Nvidia Xavier.

**Table 1 sensors-21-07399-t001:** Various Object Detection Model Performance.

Model	Platform	FPS	Real-Time
ME R-CNN	GPU	13.3	No
MFR-CNN	GPU	6.9	No
A-Couple-Net	GPU	9.5	No
DF-SSD	GPU	11.6	No
YOLOv4	Edge AI	13.6	No
SSD	Edge AI	16.9	No

**Table 2 sensors-21-07399-t002:** Rulesets. The diagonal symmetric relationship is highlighted in bold.

	${\tilde{A}}_{K_{max}}^{b}$	b1	b2	b3	b4	b5	b6	b7	b8	b9	b10	b11	b12	b13
${\tilde{A}}_{K_{min}}^{s}$														
**s1**		**fzy13**	fzy12	fzy11	fzy10	fzy9	fzy8	fzy7	fzy6	fzy5	fzy4	fzy3	fzy2	fzy1
**s2**		fzy12	**fzy12**	fzy11	fzy10	fzy9	fzy8	fzy7	fzy6	fzy5	fzy4	fzy3	fzy2	fzy1
**s3**		fzy11	fzy11	**fzy11**	fzy10	fzy9	fzy8	fzy7	fzy6	fzy5	fzy4	fzy3	fzy2	fzy1
**s4**		fzy10	fzy10	fzy10	**fzy10**	fzy9	fzy8	fzy7	fzy6	fzy5	fzy4	fzy3	fzy2	fzy1
**s5**		fzy9	fzy9	fzy9	fzy9	**fzy9**	fzy8	fzy7	fzy6	fzy5	fzy4	fzy3	fzy2	fzy1
**s6**		fzy8	fzy8	fzy8	fzy8	fzy8	**fzy8**	fzy7	fzy6	fzy5	fzy4	fzy3	fzy2	fzy1
**s7**		fzy7	fzy7	fzy7	fzy7	fzy7	fzy7	**fzy7**	fzy6	fzy5	fzy4	fzy3	fzy2	fzy1
**s8**		fzy6	fzy6	fzy6	fzy6	fzy6	fzy6	fzy6	**fzy6**	fzy5	fzy4	fzy3	fzy2	fzy1
**s9**		fzy5	fzy5	fzy5	fzy5	fzy5	fzy5	fzy5	fzy5	**fzy5**	fzy4	fzy3	fzy2	fzy1
**s10**		fzy4	fzy4	fzy4	fzy4	fzy4	fzy4	fzy4	fzy4	fzy4	**fzy4**	fzy3	fzy2	fzy1
**s11**		fzy3	fzy3	fzy3	fzy3	fzy3	fzy3	fzy3	fzy3	fzy3	fzy3	**fzy3**	fzy2	fzy1
**s12**		fzy2	fzy2	fzy2	fzy2	fzy2	fzy2	fzy2	fzy2	fzy2	fzy2	fzy2	**fzy2**	fzy1
**s13**		fzy1	fzy1	fzy1	fzy1	fzy1	fzy1	fzy1	fzy1	fzy1	fzy1	fzy1	fzy1	**fzy1**

**Table 3 sensors-21-07399-t003:** The Result of the Different Jump-threshold. Better results are highlighted in Bold.

Model	Trainset	Jump-Threshold	mAP	FPS (PC)	FPS (Xavier)
FGSC-SSD	07 + 12	0.08	74.0	60.6	19.1
	07 + 12	0.1	74.1	63.4	19.6
	**07 + 12**	**0.2**	**74.8**	**64.9**	**21.7**
	07 + 12	0.3	73.2	65.2	20.4

**Table 4 sensors-21-07399-t004:** The Information of FGSC Block Parameters.

Block	The Ratio of Skipped Parameters
FGSC1	16/256 (6.2%)
FGSC2	258/512 (50.4%)
FGSC3	294/512 (57.4%)
FGSC4	541/1024 (53%)
FGSC5	276/512 (54%)

**Table 5 sensors-21-07399-t005:** PASCAL VOC (07 + 12) Dataset Test Performance. The best FPS results of FGSC-SSD are highlighted in Bold.

Method	Platform	mAP	FPS
ME R-CNN	Titan Xp	72.2	13.3
MSA-DNN	Titan Xp	81.5	31.2
SSD	Titan Xp	81.7	31.5
MFR-CNN	Titan X	82.6	9.5
DF-SSD	Titan X	78.9	11.6
ACoupleNet	Titan X	83.1	6.9
SSD	Titan X	74.3	46
YOLOv3	1080 Ti	73.0	64.2
YOLOv4	1080 Ti	78.9	55.7
SSD	1080 Ti	73.7	56
**FGSC-SSD**	**1080 Ti**	73.8	**64.9**
YOLOv3	Xavier	73.0	19.3
YOLOv4	Xavier	78.9	13.6
SSD	Xavier	73.7	16.9
**FGSC-SSD**	**Xavier**	73.8	**21.7**

**Table 6 sensors-21-07399-t006:** Benchmark BIT Vehicle dataset Test Performance. The best FPS results of FGSC-SSD are highlighted in Bold.

Method	Platform	mAP	FPS
YOLOv3	1080 Ti	96.3	26.2
YOLOv4	1080 Ti	95.1	25.3
SSD	1080 Ti	91.4	23.9
**FGSC-SSD**	**1080 Ti**	95.5	**26.8**
YOLOv3	Xavier	96.3	20.2
YOLOv4	Xavier	95.1	17.6
SSD	Xavier	91.4	15.4
**FGSC-SSD**	**Xavier**	95.5	**21.7**

**Table 7 sensors-21-07399-t007:** Test Data Accuracy Results.

Method	Car	Bus	Truck	mAP
SSD	86.8	85.3	85.1	85.7%
FGSC-SSD	96.7	95.5	94.5	95.5%

**Table 8 sensors-21-07399-t008:** Practical Application Average FPS Results.

Method	Platform	FPS (Video 640 × 480)
SSD	Xavier	33.6
FGSC-SSD	Xavier	38.4

**Table 9 sensors-21-07399-t009:** The Vehicle Class Counting Result from Actual video.

Vehicle Types	Vehicle Numbers	Correctly Counting	Error Counting	Correctly Counted
Car	61	59	2	96.7%
Bus	44	42	2	95.5%
Truck	36	34	2	94.5%

## Data Availability

Publicly available [37] PASCAL VOC (07 + 12) and Benchmark BIT Vehicle [41] open sources datasets were analyzed in this study. These data Sets can be found here: https://paperswithcode.com/sota/active-object-detection-on-pascal-voc-07-12 (accessed on 2 August 2021) and BIT-Vehicle Dataset-Vehicle Model Identification Data Set-Programmer Sought.

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
