# Peer review of "FGSC: Fuzzy Guided Scale Choice SSD Model for Edge AI Design on Real-Time Vehicle Detection and Class Counting"

_sensors, 2021, doi:10.3390/s21217399_

Round 1

Reviewer 1 Report

A few additional and introductory notes on fuzzy logic would have been necessary

The method is based on the combination of FGSC modules to the SSD model to make detection of vehicles on the road. The method is adequately expressed, but it lacks to specify in depth the motivations for some choices, such as why choose SSD model and not YoLo (v3 or v4) as a model on which to apply fuzzy logic and then also present alternative methods for fuzzication and defuzzication.

Section 4 must be expanded because in its current state it lacks of adequate comparison with other other methods. The other experiments in section 5 have many baselines, but the text sometimes does not correspond to the value in the table. E.g. Table 6

Since experiments do not show proper comparisons with other methods, the conclusions drawn by the authors can be improved by increasing the comparison

Reviewer 2 Report

Dear Authors,

in the first part, the authors describe computer vehicle counting systems, and rightly note their superiority over manual systems. I propose to supplement this part with 2 references:

https://doi.org/10.1016/j.trpro.2021.02.038

10.1088/1757-899X/710/1/012041

In the second chapter, the authors present works related to their subject matter, I consider it good, but I suggest that the authors refer to it also in the discussion section, which is quite poorly described.

On what basis was the given FGSC-SSD neural network selected?
Correct the description of Table 2.
There are often double spaces between words in the edit layer. Some paragraphs have wrong line spacing (for example line: 382-84).

Overall the work is well written, the structure is good, but the Discussion section is too poor to be improved. Therefore, I believe that the paper, after introducing minor revission, is suitable for publication in the Sensors journal.

Best regards.
